# Noise leads to the perceived increase in evolutionary rates over short time scales

**Brian C. O'Meara**[1]*, **Jeremy M. Beaulieu**[2]*

**1** Department of Ecology and Evolutionary Biology, University of Tennessee; Knoxville, Tennessee, United States of America, **2** Department of Biological Sciences, University of Arkansas; Fayetteville, Arkansas, United States of America

\* bomeara@utk.edu (BCO); jmbeauli@uark.edu (JMB)

## Abstract

Across a variety of biological datasets, from genomes to conservation to the fossil record, evolutionary rates appear to increase toward the present or over short time scales. This has long been seen as an indication of processes operating differently at different time scales, even potentially as an indicator of a need for new theory connecting macroevolution and microevolution. Here we introduce a set of models that assess the relationship between rate and time and demonstrate that these patterns are statistical artifacts of time-independent errors present across ecological and evolutionary datasets, which produce hyperbolic patterns of rates through time. We show that plotting a noisy numerator divided by time versus time leads to the observed hyperbolic pattern; in fact, randomizing the amount of change over time generates patterns functionally identical to observed patterns. Ignoring errors can not only obscure true patterns but create novel patterns that have long misled scientists.

## Author summary

For decades, evolutionary biologists have observed that rates of evolution seem to accelerate over short time periods, a pattern seen across diverse data sources, from genomes to the fossil record. This observation has sparked debates about its implications for understanding the link between microevolution and macroevolution. Our research challenges this widely accepted notion, revealing that these apparent patterns are actually statistical artifacts resulting from time-independent "noise". By employing a novel statistical approach, we found that this time-independent noise, often overlooked as inconsequential, create a misleading hyperbolic pattern, making it seem like evolutionary rates increase over shorter time frames when, in fact, they do not. In other words, our findings suggest that smaller, younger clades appear to evolve faster not due to intrinsic properties but because of statistical noise. Ultimately, our study underscores the critical importance of accounting for inherent biases and errors in interpreting biodiversity patterns across both shallow and deep time scales.

**Data Availability Statement:** All scripts and substantial outputs are available at https://github.com/bomeara/hyperbolic_rates; DOI 10.5281/zenodo.13372718. The R package developed for

this project, hyperr8, is at https://github.com/bomeara/hyperr8.

**Funding:** This study was funded by grants from the National Science Foundation (grant DEB−1916539 awarded to B.C.O. and grant DEB−1916558 awarded to J.M.B.). The funders had no role in study design, data collection and analysis, decision to publish, or preparation of the manuscript.

**Competing interests:** The authors have declared that no competing interests exist.

## Introduction

Biology is characterized by diversity: how a modern moss survives and evolves is very different from how Cambrian trilobites did the same. Biologists thus take immediate notice when broad patterns manifest across a diverse set of lineages. One such recurring pattern is that evolutionary rates exhibit an exponential increase towards the present or over shorter time scales, highlighting a potentially new fundamental principle in how life evolves. The consistency of this pattern across dimensions of diversity, from genomes to the fossil record [1–6], highlight its universality and arguably point to the need for new conceptual bridges connecting disparate timescales of evolutionary change [7,8], despite past work showing potential artifactual causes [9]. It also challenges the long-held view that the processes playing out in the past generally behave similarly in the present—that is, life seems to evolve "faster" now. Understanding this pattern offers the potential for new insights into the underlying mechanisms that shape biodiversity.

Here we introduce a set of novel models that assess the relationship between rate and time that includes hyperbolic and linear functions of time. We show that one potentially crucial but largely overlooked factor (but see [10,11]) affecting rate patterns is the impact of empirical errors inherent in rate estimates, and that this apparently has driven the pattern observed across extinction risk, trait evolution, and diversification rate estimates over time, based on model fitting and new randomization tests. We suspect that questions about rates changing over time scales can only be properly examined, across a variety of fields, once the biasing effect of uncertainty is taken directly into consideration.

### Impish problems

The concern over spurious correlations between ratios and shared factors has a longstanding history in the statistical literature, dating back to Pearson's pioneering work more than a century ago [12]. Pearson illustrated this problem by recounting a biologist's study of skeletal measurements, specifically femur and tibia length normalized as fractions of the humerus length. Expecting a high correlation to validate correct groupings, the biologist was unaware that an "imp" had randomly shuffled bones between specimens. Surprisingly, the high correlation would persist even after this randomization, revealing that such spurious relationships arise from inherent properties of the variables rather than indicating meaningful biological connections [12–15].

Similarly, when plotting an evolutionary rate against its corresponding denominator, time, the representation becomes a plot of time against its reciprocal (i.e., k/time vs time), resulting in a relationship that is negatively biased [5,6]. In fact, if the numerator is held constant, the slope on a log-log scale must be -1.0 [1,16]. However, the numerator in an evolutionary rate, which quantifies the absolute amount of evolutionary change between two time points, is unlikely to remain constant across all timescales, as seen empirically [17]. Our hypothesis is that the observed hyperbolic pattern comes about due to this reciprocal relationship, especially given uncertainty in rates. We show mathematically why this might be the case, develop an approach for estimating the magnitude of the effect, and do randomizations and simulations to suggest that this is driving nearly all the observed empirical patterns.

### Components of evolutionary rate patterns through time

In the context of evolutionary biology, a rate is a measure of evolutionary change per unit of time,

$$r(t) = \frac{x(t)}{t} \tag{1}$$

Evolutionary change, $x(t)$, encompasses a variety of measures, including number of nucleotide substitutions in DNA [18], the number of transitions between discrete phenotypes [19], speciation and extinction events [20], or the absolute change in a continuous trait after some interval of time [21]. The specifics of the different types of evolutionary rates and how they are estimated are varied. For instance, one model may assume changes follow a Poisson distribution on a nested tree structure (e.g., substitution rates), while others may assume trait changes are drawn from a normal distribution with the rate being the measure of the variance (e.g., Brownian motion). Nevertheless, at its core, an evolutionary rate reduces to some measure of change over time.

While it is common to perceive uncertainty and error as obscuring patterns rather than contributing to them, these uncertainties and errors may be key drivers of the repeated pattern of rates increasing towards the present. For illustrative purposes, we focus on the simplest approach for calculating rates of morphological evolution to demonstrate the impact of biases and errors on the relationship between rate and time. Measurements are assembled as a set of paired comparisons that differ in some trait value (e.g., body size) measured at two separate time points, with ratio of the differences in trait and time being an estimate of the rate,

$$\hat{r}(t) = \frac{|x_2 - x_1|}{t_2 - t_1}, t_2 > t_1. \tag{2}$$

Here $x_1$ and $x_2$ are the initial and final values of the trait measurement between two time points, $t_1$ and $t_2$, respectively. Expressing a rate in such a way is similar to the "darwin," a widely used unit of evolutionary change first defined by Haldane [22]: with the darwin, the $x_i$ are the traits measured in log space.

Various phenomenological patterns can describe how rates change through time. The simplest is that $\hat{r}(t)$ is a constant rate of change, maintaining a constant value regardless of the measurement interval. For example, under a molecular clock, a mutation rate resulting from DNA copying errors is expected to be constant across clades of different ages. In other words, even though a pair of taxa sharing a common ancestor 5 million years ago are expected to have far fewer mutations than a pair sharing an ancestor 50 million years ago, on a per-time basis the rate will be identical. The biological rationale for such consistency in rate is rooted in the assumption that processes governing rates operate similarly in the past as they do in the present–a foundational premise in much of evolutionary biology. On the other hand, if mutation rates were increasing towards the present (perhaps due to a decrease in the protective ozone layer, or loss of function of mutation repair enzymes), then we would expect the rates to increase near the present.

Measurements of $x_i$ often reflect the mean for a species or even a measurement of a single individual and, therefore, represent an estimate of the true value. If each measurement estimate, $x_i$, carries some level of noise attributed to factors like finite sample size (e.g., how representative is this particular flower of the species as a whole) or measurement error (e.g., how long is this rather stretchy squid, does this messy DNA band represent an A or a T), then $\hat{x}_i = x_i + \varepsilon_i$, with $\varepsilon_i$ denoting an error component that can lead to overestimation or underestimation of the measurement. Thus, we can express the rate estimate as including error:

$$\hat{r}(t) = \frac{|(x_2 + \varepsilon_2) - (x_1 + \varepsilon_1)|}{t_2 - t_1}, t_2 > t_1. \tag{3}$$

We can reorder the terms to get:

$$\hat{r}(t) = |\left(\frac{x_2 - x_1}{t_2 - t_1}\right) + \left(\frac{\varepsilon_2 - \varepsilon_1}{t_2 - t_1}\right)|, t_2 > t_1 \tag{4}$$

The first term on the right hand-side reflects the underlying model. For example, with a constant nonzero rate the first term should be constant: larger trait differences (numerator) are balanced out by larger time intervals (denominator). However, the effect of the second fraction, the error, is quite different. Since these errors are not inherently time-dependent (e.g., sequencing errors in a DNA analysis or measurement of body lengths in dinosaurs), there are no *a priori* reasons for the magnitude of the error in a 5-million-year-old clade would necessarily be any greater or lesser than the magnitude in a 50-million-year-old clade–the numerator will come from a consistent, time independent, distribution. But this is divided by different amounts of time in the denominator. A hyperbola will invariably result from the second term when the ratio of these differences in error are scaled by time, and then plotted against time (even if the time estimate itself also has error, as it inevitably does). Thus, the overall pattern of empirical rates versus time is whatever the true pattern is plus a hyperbola coming from measurement error. At short times, under most models we expect the true difference in trait values to be small, while the uncertainty in measurements may remain high, leading to the hyperbola term dominating.

To assess the relative contribution of constant, hyperbolic, and linear functions towards a rate estimate over time, we derived a novel least-squares approach [23]. Our method allowed us to predict changes in observed evolutionary rates sampled through time by minimizing the logarithm of the residual sum of squares between the predicted and observed values. We derived a model that, at its most complex, is given as,

$$\hat{r}(t) = \frac{h}{t} + mt + b. \tag{5}$$

Here $h$ denotes the hyperbolic component [in units of $x(t)$], $m$ is scalar modulating the effect of time up and down linearly [in units of $x(t)t^{-2}$], and $b$ is a constant base rate [in units of $x(t)t^{-1}$]. Essentially, the $\frac{h}{t}$ represents the $\left(\frac{\varepsilon_2 - \varepsilon_1}{t_2 - t_1}\right)$ term in the equation above, while the $mt+b$ terms represent the $\left(\frac{x_2 - x_1}{t_2 - t_1}\right)$ term, if we are willing to assume a simple model where the underlying rate varies linearly with time (including the possibility of it being constant). The full model assumes all three components have impacted the fitted value for $\hat{r}(t)$ in some way. We also fit a range of restrictions to this model where one or more of the parameters are set to zero. For example, restricting $h = m = 0$ is a model in which the rates would be inferred to be constant through time, as one would expect from a process like a molecular clock. The maximum likelihood fit of a model was assessed using the logarithm of the residual sum of squares and first converting this into a measure of the model variance and then into a log-likelihood. To facilitate comparisons across a set of models, we converted the log-likelihood into an Akaike Information Criterion score (AIC); we also assessed confidence regions (e.g., S1–S6 Figs) around each of the parameters.

## Results

### Revisiting empirical cases of Age-Dependent Rates

We analyzed five empirical datasets that encompass diverse data types, including substitution rates (Ho *et al.* [2]), rates of body size evolution from extant and extinct taxa (measured in darwins; Gingerich [1], Uyeda *et al.* [17]), speciation rates from phylogenetic trees of extant species (Henao Diaz *et al.* [3]), and contemporary species extinction rates (Barnosky *et al.* [24]). We also included a dataset of birth rates obtained from 25,000 trees simulated under a Yule process (i.e., pure birth) with the same rate (i.e., 0.10 birth Myr$^{-1}$) and clade ages drawn from a uniform distribution. Estimates of the birth rates included a correction to account for the

biased estimates for smaller tree sizes. This simulated dataset served as a type of control given that the trees were generated without error and thus the best fit model should generally appear as a horizontal line centered on the generating rate.

In all five empirical datasets, we observed a consistent negative trend in rates through time on a log-log scale (Fig 1D, 1G, 1J, 1M and 1P). This aligns with published results indicating higher rates closer to the present, declining as time increases. When we fit our least-squares model separately to each empirical dataset, we found, with one exception, that all were best fit by a model that included all three parameters ($h$, $m$, and $b$). The Henao Diaz et al. [3] data favored a model without a base rate ($b = 0$ speciation events Myr$^{-1}$; Fig 1J), but there were three other models within 2 AIC units from the best fit model, including the full model, and all featured a positive hyperbolic parameter estimate ($\hat{h} = 1.31$–$2.75$ speciation events). One consistent finding across all five empirical datasets was that the best fit model included a positive hyperbolic parameter estimate ($\hat{h} > 0$) as well as a non-zero linear component ($\hat{m} \neq 0$). This is similar to results from De Lisle and Svennson [16], who also found that a substantial proportion of the empirical pattern could be explained by plotting a ratio versus its denominator, without specifying a particular cause. While it can be hard to estimate raw uncertainty from the empirical datasets (the data are often the observed difference in measurements without uncertainty), in two datasets there are actually multiple observations of rate for a single time point, Gingerich [1] and Uyeda et al. [17]. Using just the latter (for the former, we had to estimate the point positions, creating additional uncertainty), there were 653 time intervals where there were two or more observations of rates. If rate were a noiseless function of time, there would be no variation at a particular time; the presence of multiple rates can thus be used to estimate how much variation comes from noise, The median coefficient of variation across these intervals was 0.59, meaning the standard error was 59% the value of the mean; in 12% of the time periods, the standard error was greater than the mean.

To put these results into perspective, imagine measuring the height of the Eiffel Tower, which is accurately known to be 330 meters. If these measurements are normally distributed, centered on the true height, with a coefficient of variation of 0.59 or higher, around 5% of people might either mistakenly record the height as shorter than a giraffe or more than double the actual height. In other words, a coefficient of variation of 0.59 (or higher) represents a massive amount of variation in a measurement. However, it does not indicate how much of this the result or measurement errors or natural factors like the Eiffel Tower's height slightly changing with temperature. As Brett [25] demonstrated, when coefficients of variation are high this also increases the risk of finding misleading correlations in the data.

For the fitted rate values within each dataset, we decomposed the rate to reflect the proportion of the rate attributable to each of the three components in our least-squares model. This allowed us to assess how much of the fitted rate comes from each component as a function of time instead of relying solely on interpreting the parameters in the fitted model, the colored bands in Fig 1. Decomposing the fitted rates in this way revealed that a significant portion of the relationship between rate and time in the empirical datasets is caused by the hyperbolic component (dark pink in Fig 1D–1R). This occurred despite model fits indicating that there was signal for both constant and linear components. Any influence of other components only came with long time intervals: exactly what would be expected if there is largely time-independent noise in measurements that only becomes outweighed with signal when the true difference between the measurements is high.

One question is how this problem can be addressed. We performed simulation approaches to assess how well our approach could work at estimating the true underlying rates when the presumed error was removed (S11, S12, and S13 Figs). Our approach outperformed DeLisle

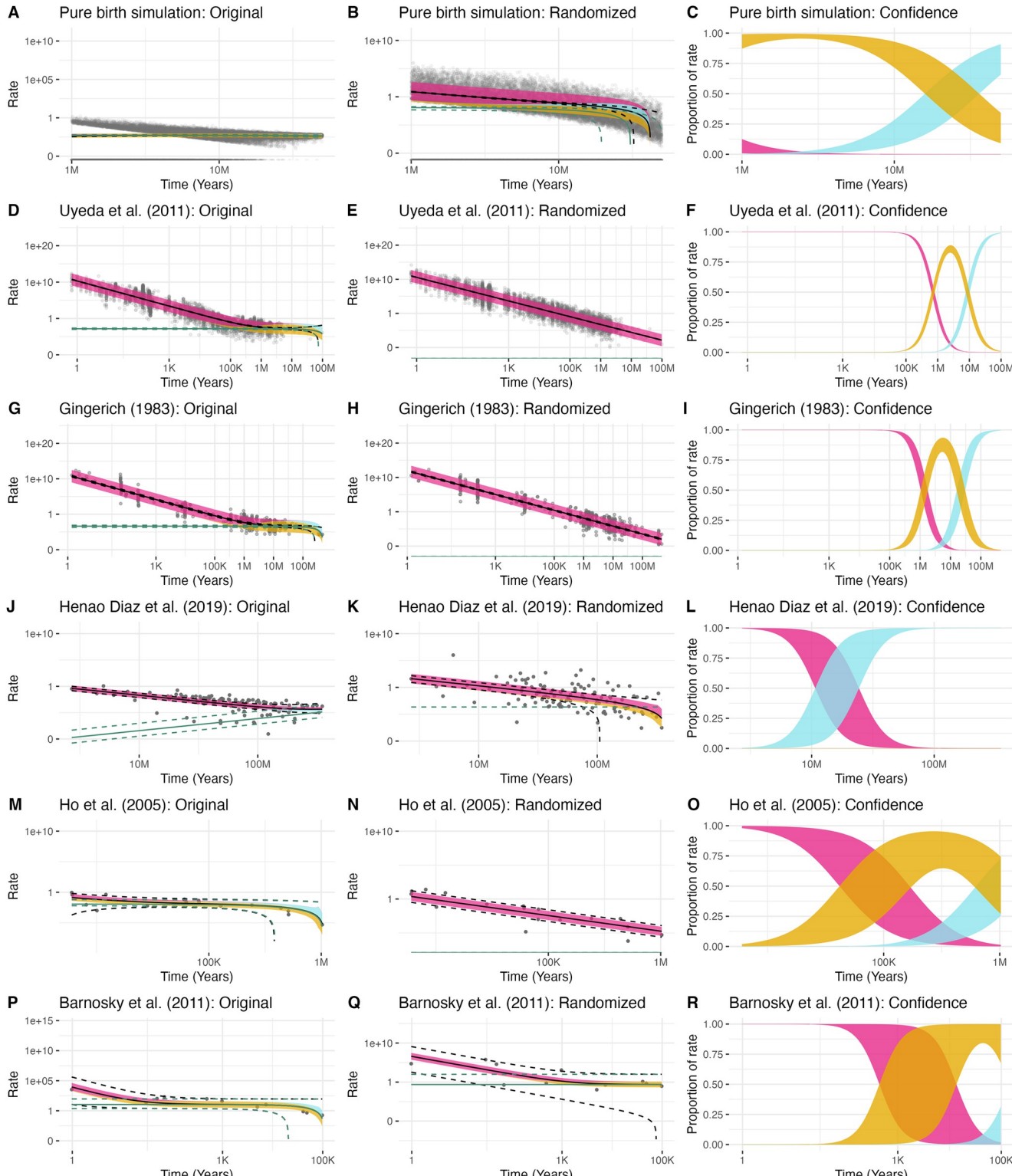

**Fig 1. Comparisons of log-linear trends in evolutionary rates.** The first column of figures is for the original data; the second column is using the randomization procedure; third shows the confidence in the assignment of rate to parameters. Time is plotted along the horizontal axis, and rate along the vertical axis, both on log scales. The first row of figures comes from a pure birth simulation, a variable process but where all information is known precisely, while the other rows come from empirical datasets. Points represent observed rate and time pairs: for example, a comparison between two populations measured at different times. For the pure birth simulation there are some rates that are zero, which are correctly placed at negative infinity on a log scale but

appear at the bottom of the plot here. The black solid and dashed lines represent the prediction and 95% confidence intervals, respectively, from the best-fitting model using our framework that estimates the hyperbolic (*h*), linear slope (*m*), and *y*-intercept (*b*) parameters, the *hmb* model. Note the close match between the predictions from the *hmb* model and the empirical data, even where the point distribution flattens out in D and G. The green lines show the prediction from the best *hmb* model of what the underlying rates would be in the absence of measurement error (deleting the *h/t* term). The thick line follows the black prediction line; colors indicate the relative impact of the various components from our least-squares model on the overall rate in the best fitting model. Dark pink represents the hyperbolic component (which appears linear on this log-log plot), goldenrod denotes a constant rate component, and blue signifies a component that changes linearly with time. For example, if the prediction for abs(*h/t*) at a particular time was 0.6, the prediction for abs(*m\*t*) was 0.3, and the prediction for abs(*b*) was 0, the height of the band at that time point would be 2/3 pink, 1/3 goldenrod. Finally, the third column illustrates the proportion of the rate from each model component over time, with band thickness indicating uncertainty in that proportion. Essentially this is the same as the color bands in the first column, but with the *y*-axis showing the proportion of the overall rate coming from each of the three parameters in the model, and band thickness showing the 95% confidence interval in how much weight each component provides (i.e., the point estimate might be that the band should be 67% pink, but the confidence interval could go from 40% to 84% pink). For the simulation in the first row, a constant rate drives the reconstructed rates (which is consistent with the generating model used), while for the five empirical plots (Fig 1F, 1I, 1L, 1O, 1R) at young ages the hyperbolic component comprises much of the rate, while at deeper times other components may become important, albeit often with substantial uncertainty.

and Svennson's [16] approach of comparing linear regression slopes to -1, as it more accurately identified trend directions. However, our method sometimes incorrectly detected trends when the generating rate was perfectly constant, especially under high noise conditions (e.g., S12 Fig, rates of 0.1 or 1, left two columns). Given the apparent large uncertainty in empirical datasets, even in cases where, say, the trend parameter was nonzero [for example, the value of -0.00007 (CI -0.00009: -0.00005) for *m* for Gingerich [1]], we would avoid drawing strong biological conclusions from this. We are somewhat more confident in estimates for the *y* intercept of the rate. For example, having a rate of 0.059 (CI 0.051–0.069) for Uyeda et al. [17] is reasonable, especially as we expect there to be a positive rate of overall evolution.

## Deriving an appropriate null expectation

A potential explanation for the interest in the severe empirical decline in rates over time is that it conflicts with the *de facto* null scientists often envision, which is a constant rate across time. One way to construct a null is to reshuffle the data and plot rates randomly with respect to times and observe whether they create a horizontal line. However, the actual data are the differences in trait measurements and the differences in time. To generate an appropriate empirical null, we developed a procedure similar to that of Sheets and Mitchell [8]. We first separated each empirical rate into its numerator and denominator components. The numerator was obtained either directly from the original data or by multiplying the empirical rate by their respective denominator of measured time. We then shuffled the numerators at random across a given empirical dataset and divided them by the unshuffled times to obtain a new set of rates and times. This not only follows the long tradition of using reshuffling to get null distributions when examining ratios against common factors [12], but also has the advantage of keeping the same distributions of times and amount of change as the original data, which can matter for some of the biases here [13]. Our expectation was that the observed relationships between rate and time will differ from results where evolutionary change and time are completely random with respect to one another.

As expected, when we randomized the numerator relative to the denominator of the empirical rates, the relationship between rate and time was almost entirely driven by the hyperbolic component across all examined datasets (Figs 1E, 1H, 1K, 1N, 1Q and 2, OR comparison type). However, we also observed surprisingly substantial similarity between the randomized and empirical results (for example, between Fig 1G and 1H). They were so similar, in fact, that when model parameters were optimized on the randomized data and applied to the corresponding empirical data, the resulting coefficient of determination ($R^2$), which indicates explanatory power, was nearly identical or sometimes greater than that of the actual empirical

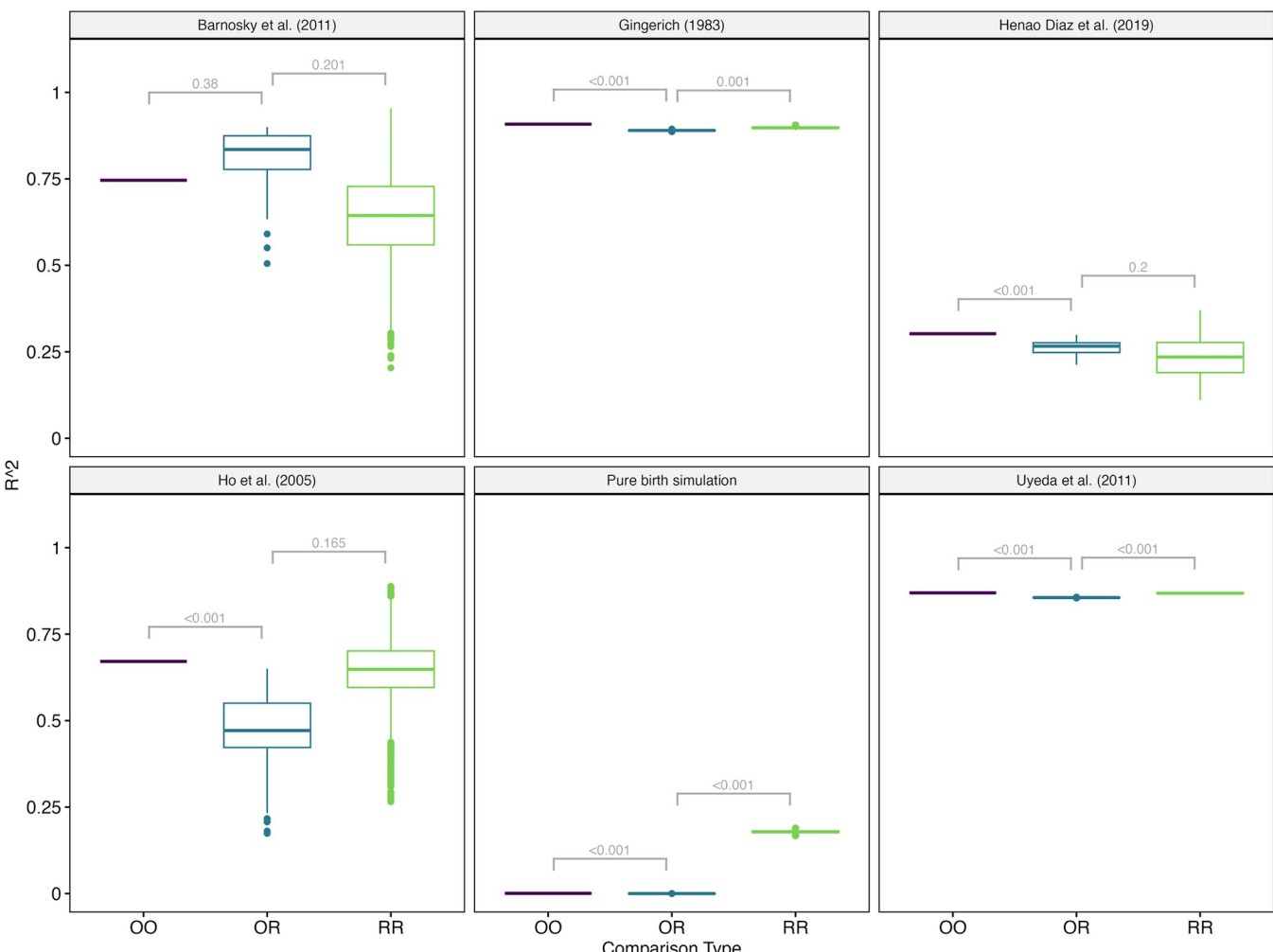

**Fig 2. Comparisons of coefficient of determination ($R^2$) between the observed fit and fits from randomizing the numerator relative to the denominator of the empirical rates.** First, we obtained the $R^2$ of the empirical original (O) data fit to the original (O) parameters (comparison type OO). Essentially, we see how well the *hmb* model with fit from the original empirical data predicts the variation in the observed rates. Second, we see how well variation in observed rates (O) is predicted when we fit the *hmb* model to randomized data (R), where presumably any true trend is erased (the OR comparisons). Randomization involved first separating each empirical rate into its numerator and denominator components, shuffling the numerators at random across a given empirical dataset, and then divided them by the unshuffled times to obtain a new set of rates and times, repeated 100 times. The last comparison examines how well randomized data is predicted from an *hmb* model fit to a different randomized dataset (the RR comparison). The expectation for data is that the best prediction (highest $R^2$) should be OO: the data one is attempting to predict are used to fit the model. Using randomized data as a prediction for the original empirical data (OR) should be poor: with randomized data, any sort of biological trend has been erased. This was often not the case. The boxplots show the distribution of the comparisons used (only one per dataset for OO, but more for OR and RR as they have randomized datasets); the brackets show a *p*-value comparing the distributions (see Methods for the details of the calculation, which avoid issues of differential power coming from different sample sizes).

fit (Fig 2). For example, in our largest empirical dataset, Uyeda *et al.* [17] with 5,886 observations, we predicted 86.9% of the variation using an *hmb* model fit to the actual data, but fitting to randomized data explains on average 85.6% of the variation (S1 Table). The fits are significantly different ($p < 0.001$), but the amount of variation explained functionally remains very similar. The fit of original data to randomized data (OR) and the fit of randomized data to different randomized data (RR) was also very similar, further suggesting that the original data functions similarly to data where amount of change and time are completely shuffled. For Ho et. al. [2], the amount of empirical variation explained by the randomized data was significantly and substantially lower than when fitted using the original data, but still remained fairly

high. With the pure birth simulation, there is no true time-correlated component of the variation (as the true rate is constant) and so the $R^2$ is extremely but expectedly low.

We hasten to acknowledge that some differences in the $R^2$ among empirical and randomized fits is expected given the inferred shifts in the contribution of three components through time. For example, while the empirical dataset of Uyeda *et al.* [17] is largely consistent with a hyperbolic distribution (Fig 1D and 1F), the rates nearer the maximum age are largely driven by the linear component—that is, the effect of $\left(\frac{\varepsilon_2 - \varepsilon_1}{t_2 - t_1}\right)$ is less than that of $\left(\frac{x_2 - x_1}{t_2 - t_1}\right)$ at deep times. This is also demonstrated in the different pattern between Fig 1D, with the original data, and Fig 1E, with the randomized data: the two lines appear very similar until deeper time points. Nevertheless, as a practical matter, the fact that we can take measurements like amount of change within *Daphnia* over a summer [26], the amount of change between late Cretaceous *Tyrannosaurus* and Triassic *Coleophysis* over 98 million years [27], and many datasets in between, shuffle these values, and have the model fit nearly as well on the original data— including explaining over 80% of the variation—suggests that the apparent patterns arise from fitting noise. By contrast, the Yule dataset (Fig 1A–1C), where there is variance due to the model (a given simulation condition can create trees of very different sizes) but no error in the number of taxa used, the tree, or the total height. In this case, the empirical pattern was very different from a hyperbola (Fig 1A) but does become hyperbolic when randomized (Fig 1B). Even an error creating a bias, such as censoring the data with zero rates without using a correction, creates a pattern very different from the hyperbola as a consequence of noise (S7 Fig).

## Discussion

Our study demonstrates that the pattern of increased evolutionary rates towards the present, observed across various axes of biodiversity, such as contemporary extinction rates [24] and macroevolutionary rates in morphology and molecular studies [1,2,17], closely resembles what we would expect from purely random data. Even within large datasets, the observed hyperbolic pattern closely mirrors noise, except at deeper times where there may be enough signal to overcome the hyperbolic pattern from noise. Furthermore, it is worth noting that while our focus was on ecological and evolutionary datasets, similar conflicts between rates measured over different time scales exist in other fields, including the Hubble tension regarding the cosmic expansion rate [28,29].

Our findings do not imply that assessing patterns of rates over time is impossible. For instance, in dendrochronology, annual tree growth is often estimated using cores with a relatively consistent number of years, rather than increasing the number of years for older samples. A recent study on contemporary bird extinction dynamics [30] employed 100-year rolling windows to estimate average extinction rates over time, and contrary to expectation, did not find the highest rate at present. Approaches like these, which maintain consistent time intervals, should likely be robust to the statistical artifacts uncovered here. However, the traditional approach of simply taking rates and plotting versus times is fundamentally flawed and practically uninformative, even with corrections. One could easily add more complexity to our linear *hmb* model, for example (such as different rates in different discrete times, rates correlating with some external variable, and more), but at least with the amount of variation we see in the empirical datasets here we would not believe any results are biologically informative. Moreover, though our envisioned noise mechanism alone generates data nearly indistinguishable from empirical data, it is far from the only bias or source of error in these data (see also [31]). First, it is worth reflecting on the various kinds of causes for noise at the tips. For example, there is simple uncertainty (a ruler with finite precision), traits that are themselves somewhat ambiguous (width of a jellyfish), sampling uncertainty, or phenotypic plasticity.

Estimates of time can be biased and associated with large amounts of uncertainty. There is also substantial ascertainment bias in which datasets scientists examine. Over short time periods, we only look at cases where we expect a detectable amount of change; over long time periods, we only compare things similar enough to make comparison sensible (e.g., body length between two species of dinosaurs, not body length between a dinosaur and a tree). For diversification analyses, clades below a certain size are often excluded, creating a bias in recovered rates, especially for clades originating more recently. Finally, models themselves can be wrong, which can result in incorrect estimates of amount of change, time, or the interplay between them.

An implication of our results is that any errors present in measurements of evolutionary change not only greatly affect rate estimation overall, but also any comparative test that examines rate shifts within a single phylogenetic tree. Consider a scenario where we want to compare evolutionary rates between a paraphyletic group of taxa and clade within it (such as seed size evolution in nonflowering plants versus angiosperms). Any inaccuracies in trait measurements or character state assignments within the tree could lead to the erroneous inference that the subclade has a higher rate, simply due to less overall time in terms of branch lengths to mitigate the effects of these errors (S10 Fig). This discrepancy also creates a concerning disconnect between the expected error rates for specific phylogenetic comparative models, derived from simulations, and their actual behavior in empirical settings when assessing rate differences within and among groups. That is, null models for many comparative tests may fit far worse in an empirical setting than might be indicated by careful simulation study conducted in the absence of measurement error [32], providing alternative explanations with artificially inflated support. Even with Bayesian approaches that attempt to integrate over uncertainty in results, data are still typically assumed to be free of errors, and this being incorrect will tend to lead to increased rates, especially where the magnitude of the effect of the error is high relative to the magnitude of the effect of the signal. This may also pose substantial issues to molecular and fossil dating analyses.

We anticipate that our findings will encourage further development of approaches that acknowledge measurement error when estimating rates. In models of continuous trait evolution, measurement error can be addressed by directly inputting the standard error of the species means for a trait to the model [33]. However, this practice is not yet standard, but at the very least, measurement error could be incorporated as a parameter to estimate in the model. The current standard approach forces it to be zero, which is a very strong, and likely very inaccurate, assumption. In other comparative models, especially those related to the evolution of discrete characters (e.g., higher-level phenotypic characters exhibiting a finite state space), rates are determined without methods to mitigate potential errors, like inaccuracies in character state assignments. Several approaches derived for dealing with sequence error in phylogenetic inference could also prove useful in this context (see [34, 35]) Such errors have a clear impact of inflating rates over shorter time scales (e.g., Figs 1 and S9). More generally, this points to the need to incorporate the potential effects of measurement error and other uncertainty in causing a pattern, not just noise. This has been dismissed as a potential cause of the pattern here [8] but both randomization and simulation presented here show how the empirical pattern can come directly from error in the numerator.

One unexplored issue is that while our study solely focused on errors in measurements of evolutionary change, errors in the temporal component underlying a rate can also significantly influence rate estimates. In certain scenarios, such as diversification models, we suspect errors in branch lengths might counteract the observed patterns, at least partially. Therefore, given the potential biases in molecular age estimates [36–38], it is prudent to prioritize the development of integrative approaches that address errors not only in measurements of evolutionary change, but also in the underlying branch lengths.

Finally, our findings demonstrate the importance of using informed nulls as expectations of a process [39], especially as errors have a biased effect on rate estimates. Understanding how rates vary both within and among phylogenies will always remain a key part of biology, as they can point to important principles governing biodiversity across disparate timescales. However, statistical artifacts can still impishly affect our results, particularly when we dismiss uncertainties in our measurements as inconsequential.

## Materials and methods

### Overall workflow

Other than the Yule simulations themselves (see below), all analyses were done in R 4.4.0 [40] using *targets* [41]. This creates a reproducible workflow that allows for caching intermediate results. All scripts and substantial outputs are available at https://github.com/bomeara/hyperbolic_rates; doi: 10.5281/zenodo.13372718. The R package developed for this project, *hyperr8*, is at https://github.com/bomeara/hyperr8. Other packages used included *ggplot* [42], *ape* [43], *dplyr* [44], *Rmpfr* [45], *tidyr* [46], *nloptr* [47], *dentist* [48], and *scales* [49].

### Empirical data and Yule simulation

We fit our models to five empirical data sets and rates generated from a simulation of Yule trees. We used the body size dataset compiled by Uyeda *et al*. ([17], their "Dryad7.csv"), which integrates contemporary field studies, historical field data, and the fossil record for mammals, squamates, and birds. This was filtered for points that were BodySizeCorrelated == 1, but no further filtering was done. Due to its historical importance, we also used the morphological rate data set of Gingerich [1]. We used the mitochondrial substitution rate (measured in sites $Myr^{-1}$) data of Ho *et al*. [2]. Finally, we used the rates of contemporary species extinction (E/Msy, or extinctions per million species-years) from Barnosky *et al*. [24].

The original study of Henao Diaz *et al*. [3] estimated diversification rates (i.e., speciation, extinction, and net diversification rates) from these trees. However, these were not available (Matt Pennell, personal communication). We therefore re-estimated diversification rates by fitting a constant birth-death model to each tree using modified functions extracted from the R package *TreePar* [50]. Specifically, we re-parameterized the likelihood function to search for the maximum likelihood estimates of turnover, $\tau$, which is speciation + extinction, and extinction fraction, $\varepsilon$, which reflects the ratio of extinction rate to speciation rate. We conditioned the likelihood on survival (i.e., probability that the observed tree could have gone extinct by time, $t$). For each tree, we transformed estimates of $\tau$ and $\varepsilon$ back to the original speciation and extinction variables.

In the case of Gingerich [1], Ho *et al*. [2], and Barnosky *et al*. [24] the original data were not readily available. We therefore used plotdigitizer.com to manually digitize the points from the log-log plots, as they allowed easiest visibility of individual points. For Ho *et al*. [2] we recreated Fig 1*A* from [7]. For Gingerich [1], in cases where there were multiple data points per symbol, and so we recorded one point for each replicate. For example, a symbol of "5" became five points at that location, whereas an "X" means 10 or more. In the latter case, we encoded these as ten points.

For the Yule simulated rates, we generated a vector of 25,000 ages from a uniform distribution, starting at log(1 Myr) and ending at log(50 Myr). We then transformed the ages back to their original units. This ensured that the sampling was approximately even on a log-log plot. We then simulated a tree at each of these ages in the R package *TreeSim* [51] with a known birth rate of $\lambda = 0.10$ births $Myr^{-1}$. Each simulation began with two lineages and terminated once the tree reached the specified age. Estimates of the birth rate were obtained analytically

using our newly derived unbiased estimator, $\hat{\lambda}^*$, for Yule birth rates (see section *Deriving an unbiased estimator for the Yule birth rate* section below).

## Least-squares model

We implemented a set of least-squares models into an R package, *hyperr8*, whose namesake is based on our expectation of hyperbolas and as an homage to Mike Sanderson's *r8s* software [52]. The basic idea is to minimize the logarithm of the residual sum of squares between the log predicted and log observed rates coming from a model which, in its most general form, predicts a rate estimate as:

$$\hat{r}(t) = \frac{h}{t} + mt + b. \tag{6}$$

This general model is referred to as the *hmb* model after its free parameters *h*, *m*, and *b* that correspond to the hyperbolic, linear, and constant components. We can fix any of the parameters to 0 to create simpler models nested within this general one. For example, *h0b* fixes *m* at 0 and estimates *h* and *b*. The simplest model in the set of possible models restricts $h = m = 0$ and the rates are assumed constant through time. To avoid some extreme rates driving the fit, we used the log of the empirical rates and the log of the predicted rates when calculating the residual sum of squares. If any empirical data set rates contain rates of zero, as was true in our pure birth simulation for trees that started and ended with two taxa, we used a log1p transform on all rates. This is done automatically in the *hyperr8* software.

For each empirical data set we fit and compare a set of models that range in complexity, as well as assess uncertainty in their inferred model parameters. We accomplished this by taking advantage of the fact that one can convert the residual sum of squares to a log-likelihood. We first calculated the maximum likelihood estimate for the variance of the residuals, $\hat{\sigma^2}$, by dividing the residual sum of squares by the number of observations. The log-likelihood is then calculated as,

$$log(L) = -.5n*log(\hat{\sigma^2}), \tag{7}$$

and we maximize this to obtain the MLE of model parameters for a given model. The log-likelihood and the number of free parameters were also used to compute the Akaike Information Criterion ([53], AIC). This allowed us to use *dentist* [48] to compute uncertainty around the parameter estimates (see *Adding uncertainty in our plots* section below).

Our optimization procedure starts with a diverse set of starting points and algorithms from the *nloptr* package [47] in R, to fine-tune a given model. Initially, we optimize model parameters using fixed starting points with the NLOPT_LN_SBPLX [54] algorithm. Subsequently, a different set of fixed starting points is applied to the same algorithm, followed by random starting points centered on the best values found in the first round of optimization with NLOPT_LN_SBPLX. The optimization process cycles through the NLOPT_LN_BOBYQA [55], NLOPT_LN_SBPLX, and NLOPT_LN_NEWUOA_BOUND [56] algorithms to mitigate the risk of suboptimal fits. If a better log-likelihood is discovered during the parameter search, those parameters and log-likelihood are used for comparison with other models. This search effort is applied to both the original data and each of the randomized datasets. The resulting fits are provided in S1 Table.

## Contribution of each parameter to the fitted rate

Using the best-fitting model, we calculated the values of $h/t$, $m*t$, and $b$ for any given value of $t$. Since these are usually summed to obtain the fitted rate, $\hat{r}(t)$, it is exceedingly rare that a fitted

rate will be estimated as being less than zero. For example, the rate could decrease with time when $m$ is less than one, but a high enough value for $b$ would ensure that $m*t + b$ remains above zero over the range of times examined. Nevertheless, we took the absolute value of each component anyway when calculating its overall contribution to the rate. If $h = 2$, $m = -1$, and $b = 0$, then at time $t = 1$ the overall rate is $2/1 + -1*1 + 0 = 1$, but the contribution of the linear component, $m$, is $|-1| / (|2|+|-1|) = \frac{1}{3}$, while at time $t = 0.5$ the overall rate is $2/0.5 + -1*0.5 + 0 = 3.5$ and the contribution $m$ is $|-0.5| / (|4| + |-0.5|) = \frac{1}{9}$.

The rightmost column in Fig 1 of the main text shows the predicted rate as a band, but we also wanted the color of that band to communicate the relative contribution of each component. This was accomplished by using *geom_ribbon* within *ggplot2* [42] with ribbon widths proportional to the contribution of each component. However, this visually makes the ribbons look narrower when the rate has a slope with high magnitude, so we estimated what the slope is locally, which was then used to inflate the ribbon width on steeper parts of the predicted rate line.

## Adding uncertainty in our plots

Overall uncertainty in the rate estimates as we all as individual estimates of $h$, $m$, and $b$ were inferred using *dentist* [48]. Functions within *dentist* work by "denting" the likelihood surface, trying sample points around the rim to approximate the confidence interval one would calculate from a chi-square distribution with the same number of degrees of freedom (e.g., a difference in likelihood of 1.92 log likelihood units for a single parameter). In some ways, this is similar to sampling a credibility interval in a Bayesian analysis, but without the possibility of rescuing a ridge in likelihood space through the use of priors. This gives us a set of points inside a region that has "good enough" likelihood. For well-behaved surfaces with a unimodal peak, this will look approximately like an ellipse when any pair of parameters are plotted. To be conservative, we take the minimum and maximum of each parameter inside the confidence region (i.e., the highest and lowest values for $h$, $m$, and $b$) and use these to calculate the range of possible values for the component contributions. For example, we compute the contributions of hyperbolic, linear, and constant components for $h_{max}$, $m_{max}$, $b_{max}$, then do it again for $h_{max}$, $m_{max}$, $b_{min}$, then for $h_{max}$, $m_{min}$, $b_{min}$, and so forth until we have tried all eight combinations at a given time point. The right-hand column of Fig 1 in the main text was created in this way, where the vertical thickness of the bands is due to their range of uncertainty in their proportions. We used *dentist*'s defaults for estimating the likelihood width to use as well as number of steps. For figures on the surfaces, we increased the number of steps to 10,000 and reran the analyses to create denser plots that are easier to see. The results from the dentist for each data set are shown in S1–S6 Figs.

## Randomization procedure

As described in the main text, we used a randomization procedure to generate a new empirical null to compare against our empirical fits. This involved first separating each empirical rate into its numerator and denominator components. Some of the datasets included information on the numerator (e.g., difference in log body size for datasets whose rates are measured in darwins) and the denominator (e.g., time separating the pair of values being compared), while others included only rate and time. For the latter, we imputed the numerator by multiplying the rate by time. We then shuffled the numerators at random across a given empirical data set, and then divided them by the unshuffled times to obtain a new set of rates and times. This was repeated 100 times for each dataset.

Rates derived solely from phylogenetic trees, such as those in the Henao Diaz *et al.* [3] dataset and our pure birth simulations, present a unique challenge for our randomization

procedure. Typically, rates are assessed across the entire tree, which reflects the total time span of all branch lengths. However, the time indicated on the *x*-axis typically represents the age of the tree, which, although related [57], are far from perfectly correlated. Utilizing the age of the tree to calculate the numerator would significantly underestimate the rate. In such cases, we opted to use the total branch length of the tree to calculate both the numerator and the denominator when recalculating the rate, while still employing the tree's age for the *x*-axis and model fitting. We do note that for the Yule simulation dataset, we did re-run all analyses using total tree height as opposed to clade age and did not find any substantive and qualitative differences from the results presented in the main text (S7 Fig).

## Comparing empirical and randomized data sets

We commonly use statistical approaches like bootstrapping, jackknifing, or randomizations to estimate parameters from resampled data as a means of testing for robustness. Here, we took an additional step by evaluating the predictive accuracy of these parameters derived from resampled data on the original empirical data. The first step is assessing how well the best model fitted from the empirical data predicts the empirical rates. This is done by calculating the coefficient of determination ($R^2$) between the empirical rates and predictions from the fitted model (using log rates for the calculation, with the exception of the pure birth simulation where we used log1p given the presence of rates of zero). In Fig 2, this is OO, the comparison of the empirical original (O) data fit to the original (O) parameters. We also compared the fit of the empirical original data to predictions from the randomized data (OR). This allowed us to see how much the random data "null" predicts the original data. It is possible that, even if the original data were pure noise, the fit would still be worse when using a different randomized dataset since it was not tuned to the focal dataset. To check this, we also computed the $R^2$ between a randomized dataset and the fitted values from a different randomized dataset (the RR column). Our expectation was that if the reshuffled data substantially differed from the empirical data, the fitted values from the reshuffled data would explain little, if any, of the variation in the empirical rate. By contrast, if the empirical pattern is largely noise, as we suspected, there will be little difference between the datasets.

We also sought to highlight any significant differences between the original data to original fit (OO) to original data to randomized fit (OR), and the OR fit to the randomized data with different randomized data fit (RR). One issue with this is differences in sample sizes. With OO, for a given dataset and model, there is only one calculation of $R^2$; with OR, if we have 100 randomizations there are 100 values for $R^2$; with RR, since we do not fit the same randomized dataset to itself, there are (100–1)*100 = 9,900 comparisons. So if comparing OO with OR, we would be comparing 1 value with 100 to see if they come from the same distribution; when comparing OR to RR, we would be comparing 100 values with 9,900. Using a standard *t*-test or Wilcoxon rank-sum test we would expect less power to detect difference in the OO-OR comparison (1 vs 100) than in OR-RR (100 vs 9,900). This makes it more likely that we would find no significant difference between the OO and OR distributions but would find one between OR and RR distributions. This relative lack of power would artificially support our hypothesis that there is no difference between empirical and random data. To avoid this risk, we used percentiles to compute the *p*-value for the OO-OR comparison, with the *p*-value equaling 2*max(percentile, 1-percentile), where the percentile is where the OO value is relative to the distribution of OR values. In order to have a comparable value for OR-RR comparison, we took a single $R^2$ value from the set of RR, computed its percentile given the values from OR, converted this to a *p*-value as for the OO-OR comparison so sample sizes for the OO-OR and OR-RR comparisons were equal. For the OR-RR comparisons we repeated this for 1,000

random draws from the RR distribution to reduce variance in the *p*-value from choosing a single representative. The average of these was taken as the *p*-value for the OR-RR comparison.

We also stress the importance of distinguishing biological significance from statistical significance, something long emphasized in the literature [58]. In the context of our work, a *p*-value is the probability of two distributions being exactly equal. For example, in the Uyeda *et al.* [17] dataset, it is clear from Fig 1 in the main text that while the empirical dataset is largely consistent with a hyperbolic distribution, the few points of maximum age are more consistent with a linear distribution. This is reflected in Fig 2 in the main text, where the fit of the empirical data to predictions from randomized data are slightly worse than fit to empirical data predictions. Due to the large size of the dataset, there is little variance in the former, so the latter is outside the observed distribution and thus is quite significantly different. Biologically, though, an $R^2$ of 86.9% using the empirical data is not substantially better than an $R^2$ of 85.6% using randomized data. It is heartening that on a huge dataset the pattern is not identical to random, but this reflects the rates nearer the maximum age being largely driven by the linear component (see Fig 1 main text).

As a visual aid, we have also made an animated gif comparing the original and randomized datasets. Given issues with drawing rates of zero for the pure birth simulation, we arbitrarily assigned those rates to be 10% of the minimum value of the nonzero rates for ease in plotting. This is available in our supplemental documents (10.5281/zenodo.13372718).

## Deriving an unbiased estimator for the Yule birth rate

Prior to the development of our least-squares approach described above, we first simulated trees under a pure birth (Yule) process to explore potential artifacts or confounding behaviors that might lead to rates exponentially increasing towards the present when the generating rate is set to be constant through time. We generated a vector comprising 1000 clade ages, uniformly sampled in log space along a line ranging from 1 Myr to 50 Myr. We then simulated a tree at each of these ages with a known birth rate of $\lambda = 0.10$ births Myr$^{-1}$. The simulation began with two lineages and terminated once the tree reached the specified age. Estimates of the birth rate were obtained analytically using the MLE estimator of the birth rate, $\lambda$, as derived in [59],

$$\hat{\lambda} = \frac{n-2}{\sum x_i} \tag{8}$$

where denominator represents the sum of the edge lengths, $x_i$, in a tree. We then plotted all 1000 rates against time and fit a loess curve to assess the trend through time.

One immediate observation was the impact of zero rates—i.e., trees starting at $n = 2$, but which fail to bifurcate after time, $t$. In an empirical context, such estimates are almost always either discarded (i.e., censored) or not included due to the fact such clades are inherently uninteresting (evolutionary "minivans" [60]). However, not including these zero rates produces an ascertainment bias where the higher rates nearer the present, which are a result of trees going on a run of speciation events early, are not properly counterbalanced by the higher instances of zero rate trees over shorter time scales. As a result, the mean shifts up where the no change results are more frequent, which we suspect explains the exponentially increasing birth rate estimates towards the present in our Yule simulations (S8 Fig).

Indeed, when zero rate trees are included, the exponential increase disappears entirely, but instead with the overall rate estimates showing a *downward* bias towards the present. In other words, when examining birth rates across time, as we do here, after correcting for ascertainment bias (e.g., only including trees ≥3 taxa) by including the zero rate trees the trend shows

an opposite trend of rates slightly *decreasing* towards the present (S8 Fig). A regression line on the log1p transformed data showed a significant positive slope (slope = 0.013), with clade age still accounting for over 6% of the variation. Again, this is due to the higher frequency of trees that failed to speciate over shorter time intervals. When these trees are included, they effectively drive the regression line down.

The remaining downward bias suggests that the Yule birth rate, $\hat{\lambda}$, is a biased estimator of $\lambda$, which is not unusual for likelihood-based estimators. Here we quantify the remaining bias by relying heavily on the formulas from Steel and Mooers [57]. A key insight from [57] was that the expected average edge length in a tree grown under a Yule process is actually $\frac{1}{2\lambda}$, as opposed to $\frac{1}{\lambda}$ that might follow from the assumption of exponentially distributed wait times before any given lineage splits on a Yule tree. This is because each speciation event contributes to the average edge length, and each event creates two lineages, leading to the factor of 1/2 in the expression. For comprehensive mathematical proofs under various conditions (i.e., conditioning) we refer the reader to [57].

We use the expected average edge length, $\frac{1}{2\lambda}$, to express the expected value of the MLE estimator of $\lambda$ as,

$$E\left(\hat{\lambda}\right) = E\left[\frac{n-2}{\frac{(2n-2)}{2\lambda}}\right], \tag{9}$$

where the total length is defined as the expected average length $\frac{1}{2\lambda}$ multiplied by the 2$n$-2 edges. This expression simplifies to,

$$E\left(\hat{\lambda}\right) = \frac{\lambda(n-2)}{n-1}, \tag{10}$$

which we then use to derive the bias of the MLE estimate of $\hat{\lambda}$ by subtracting $\lambda$ from $E(\hat{\lambda})$:

$$Bias\left(\hat{\lambda}\right) = E\left(\hat{\lambda}\right) - \lambda = \frac{\lambda(n-2)}{n-1} - \lambda = \frac{-\lambda}{n-1}. \tag{11}$$

The expected value is indeed less than the true value and therefore provides an underestimate of $\lambda$. To correct for this, we can multiply $\hat{\lambda}$ by a "correction" multiplier of $\frac{n}{n-1}$ that converges to 1 as $n$ increases.

Note that when $n$ is small, and zero rates are not included, this correction will amplify higher rates, potentially exacerbating the trend of exponentially increasing rates toward the present. However, when zero rates are included, the rates show a remarkably constant trend through time (S8 Fig). A regression line on the log1p transformed data showed a slope that is essentially zero (slope = 0.005), with time explaining less than <1% of the variation in the birth rates ($R^2$ = 0.004). Thus, throughout the main text, we exclusively focus on estimates of $\hat{\lambda}^*$ through time.

## Yule sensitivity to excluding zero rates

As mentioned above, most diversification studies and simulations typically exclude zero rates from their analyses, which can introduce bias. Excluding zero rates fails to properly balance higher rates nearer the present with the higher instances of zero-rate trees over shorter time scales (see S7 Fig). For our least-squares model, removing zero rates from our Yule simulation dataset necessarily shifted the signal towards the hyperbolic component (S7 Fig). We were concerned that if our randomization procedure incorrectly attributed exponentially increasing rates towards the present to errors when there were none, it would also raise questions about

the empirical results, which also omitted zero rates. To test this, we reran all analyses on the Yule simulation dataset with the zero rates excluded. As expected, the strongest signal nearest the present was for the hyperbolic component (S7 Fig). In this case, the randomization creates a pattern very different from the hyperbola as a consequence of noise.

### Within and among tree rate bias

One implication of our study is that errors in measuring evolutionary changes not only greatly affect rate estimation overall, but also any comparative test that examines rate shifts within a single tree. For instance, consider a scenario where we want to compare evolutionary rates between two sister clades that differ in age. Any inaccuracies in branch lengths or character state assignments within the tree could lead to the erroneous inference that the younger clade has a higher rate, simply due to less overall time to mitigate the effects of errors.

To demonstrate this, we conducted simulations of character state transitions and estimated transition rates with and without error. We ran 1000 simulations each on two types of 64-taxon trees that represented extremes in tree balance: pectinate (i.e., "caterpillar" trees) and perfectly balanced trees, where diversity is evenly distributed between sister clades. The age of each tree was varied by scaling clade age using values from a vector of 1,000 ages drawn from a uniform distribution, ranging from log(5 Myr) to log(20 Myr). We used a simple transition model that assumed equal transition rates between states 0 and 1, and we chose generating rates of 0.01 transitions $Myr^{-1}$ for pectinate and 0.05 transitions $Myr^{-1}$ for balanced to ensure that a large number of invariant or saturated data sets were not produced across our age range. Finally, we estimated the transition rates in the R package *corHMM* [61] on the resulting data sets with and without error, with error being introduced by choosing six taxa at random and changing their true state to the opposite and incorrect one.

The results of these simulations are shown in S9 and S10 Figs. First, in the absence of error the estimated transitions rates are, on average, constant across time and are centered on the generating value. However, when we introduce error, the rates are significant and negatively correlated with time, which is consistent with findings presented in the main text. Importantly, if we were to compare the rates between a clade that is say 5-million-years-old that is sister to a clade of say million-years-old, rates be higher in the younger clade simply as a byproduct of errors. Particularly intriguing is the observation that pectinate trees tend to mitigate error impacts more effectively than balanced trees, irrespective of the generating rate (not depicted). Plotting results based on total time (i.e., the sum of all branch lengths) reveals a notable yet intuitive trend–namely, that a young pectinate tree contains significantly more time overall than a balanced tree of the same size. Thus, this underscores the challenges in accurately estimating rates within and among trees that contain errors.

### Other sources of bias

Note that the biases arising from error are unlikely to be the only biases playing a role in rate estimation. As we have written elsewhere [60], there is a likely substantial effect of human selection biases. Scientists study *Daphnia* over a summer because the rate is high enough that they can expect to find a pattern, ignoring all the other groups with multiple generations over that period but less evidence for size change. Scientists study dinosaur size differences over 98 million years because their overall similarity is high enough that a comparison seems reasonable; other groups that have changed enough over such a long span of time that they are no longer considered comparable "things" are ignored. Within groups, we focus on the ones where traits are variable: no one studies gain or loss of having skulls in mammals, nor gain or loss of woodiness in oaks, or gain or loss of powered flight in cnidarians, instead focusing on

the groups that have elevated enough rates to be practical to study. There are also biases about sizes of groups to study and the way we identify groups, taking the tens of millions of clades on the tree of life and only choosing to assign names to some of them, often seemingly based on key characters arising on the branch leading to a group. We choose to study "X-idae" because that clade received the family name, rather than one node rootward or tipward. The remarkable thing about our study is that while those biases no doubt play a role, the fact that just randomization creates a nearly identical pattern to the actual data suggests that error alone has a dramatic effect all on its own.

## Evaluating fit to simulated distributions

To assess how well the new method in the hyperr8 package performs, we generated data under a variety of scenarios. The first was a constant rate of 0.4 changes per time unit (which went from 0.01 to 50-time units). The second was a rate that started at 0.6 changes per time unit and increased with a slope of 0.01 per time unit. The third was a rate that started at 1 change per time unit and decreased with a slope of -0.01 per time unit. The fourth was a sine wave with a period of 26-time units. This is something that our method should perform poorly at, as such periodicity is not a component of the model (this model is inspired by [62]). For added realism, we added error through using times with noise (standard deviation of 10% of the mean). Ten replicate datasets of 500 and 5000 data points were simulated by taking the expected rates, multiplied them by time to get the numerator, and then sampled from a normal distribution with sd of 0.000001, 0.01, 0.1, and 1 and mean at the predicted numerator, then divided by time. Some of this variation seems quite large, but it spans the empirical data. For example, the Uyeda et al. [17] dataset had 79 time points where the inferred rate at that time point had a coefficient of variation (ratio of standard deviation to mean) greater than 1. Results are shown in S11 and S12 Figs.

## Caveats

This study's outcomes represent a conflict of interest for us, as scientists whose careers are based on extensive work developing new ways to estimate rates of evolution in a variety of contexts [61, 63–65]. Rates are crucial for understanding biology, but their usefulness is diminished, at least in the short term, by inherent time-dependence primarily caused by noise. Although we aimed for objectivity in interpreting our results, we may have inadvertently downplayed the impact of the statistical artifacts we uncovered.

There were also analysis choices that could affect communication of results. For example, we found that the parameter estimates from some of the empirical data sets were slightly outside the distributions of parameter estimates from the randomizations. This could be spun to demonstrate how significantly different the results are, but in practice the differences have little impact (as shown by the similarity in the $R^2$ results).

While we did not set a minimum or maximum dataset size, we also did not sample every published comparison of rates versus time. Instead, we limited our focus to the datasets that have been given attention recently and span a variety of rates ranging from molecules to extant species and trees to fossils [7]. Finally, we developed a robust optimization procedure into the *hyperr8s* package, starting with multiple starting points and using algorithms in *nloptr* and *dentist* to explore parameter space. However, no algorithm guarantees success, and incorrect estimates could result in poorer fits but unlikely to produce the hyperbolic patterns we discovered here.

Finally, one issue with the original datasets is non-independence of the data points. This includes the standard phylogenetic non-independence because of shared ancestry [21], but

also stems from the way some of the comparisons were calculated. For example, Uyeda *et al.* [17] compiled information from dozens of studies, leading to over 6,031 comparisons, but there are only 1,684 distinct samples used. To ensure comparability with earlier results, we did not correct for either of these issues.

## Supporting information

**S1 Fig. Overall uncertainty in the individual estimates of h, m, and b for the Yule simulation dataset.** The top row displays univariate confidence regions for each parameter fitted to the Yule simulation dataset. The best model, denoted as *hmb*, allowed all parameters to vary freely. Sampled parameter values are represented by dots, with gray dots indicating values outside the confidence region and black dots inside. The blue line indicates the boundary between these points. This representation offers a flattened view of the multidimensional analysis; within the horizontal range of black points, some gray points may exist where values for other parameters lead to poor likelihoods. The red circle marks the maximum likelihood estimate. The second row presents bivariate plots, with colors indicating confidence regions as described above. In an effective analysis, the black region in these plots should form an ellipsoid shape, surrounded by gray. To obtain a conservative estimate from the points, such as those in the rightmost column of Fig 1 in the main text, a rectangular prism was placed around the clusters of black points. The range of values, such as the proportion of hyperbolic weight, was derived from calculations at the vertices of this prism. The overall approach of dentist resembles Markov Chain Monte Carlo in Bayesian analysis but does not rely on prior distributions. Instead, it focuses on establishing bounds rather than defining a region that comprises a certain proportion of overall probability.
(TIF)

**S2 Fig. Same dentist plot as in S1 Fig but showing parameter uncertainty for the Uyeda et al. [17] dataset.** The best model was *hmb* so all parameters were free to vary.
(TIF)

**S3 Fig. Same dentist plot as in S1 Fig but showing parameter uncertainty for the Gingrich [1] dataset.** The best model was *hmb* so all parameters were free to vary.
(TIF)

**S4 Fig. Same dentist plot as in S1 Fig but showing parameter uncertainty for the Henao Diaz et al. [3] dataset.** The best model was *hm0* so the *b* parameter is fixed at zero.
(TIF)

**S5 Fig. Same dentist plot as in S1 Fig but showing parameter uncertainty for the Ho et al. [2].** The best model was *hmb* so all parameters were free to vary.
(TIF)

**S6 Fig. Same dentist plot as in S1 Fig but showing parameter uncertainty for the Barnosky et al. [24].** The best model was *hmb* so all parameters were free to vary.
(TIF)

**S7 Fig. Fig 1, but with additional ways to handle the Yule simulation dataset presented in the main text.** The first column is for the original data; the second column is using the randomization procedure. The horizontal axis is time; the vertical axis is rate (both on a log-scale). Dots show individual rate estimates; the black line shows the regression from the best fitting model and the dashed lines show the 95% confidence interval around that regression. The thick line shows the relative impact of the magnitude of each component on the overall

rate in the best fitting model: dark red is from the hyperbolic component (which would be linear on this log-log plot), goldenrod is from a constant rate component, and blue is from a linear component. Note that for the pure birth simulation there are some rates that are zero, which are placed at negative infinity on a log scale; *ggplot2* handles these by plotting them along the x-axis. The third column shows the proportion of the rate from each component over time, with the thickness of the bands representing the uncertainty in that proportion. (TIF)

**S8 Fig. Trends of the Yule birth rate through time under various conditions of the data and the estimator.** We simulated 1000 Yule trees that assumed a constant rate of 0.1 birth My-1, (dotted black line) regardless of the time. When zero rate tree are excluded (i.e., censored), and we rely on the MLE of the birth rate, $\hat{\lambda}$, we find that rates exponentially increase towards the present (blue line, MLEcensored+biased). However, when we include the zero rate trees the line dramatically shifts downwards, with rates nearer the present being lower than in deeper time frames (purple line, MLEuncensored+biased). This suggests that the MLE estimator of the birth is biased. When we apply a "correction" to account for the bias, we see that the rates are indeed constant through time (yellow line, MLEuncensored+unbiased), which is consistent with the simulation scenario. (TIF)

**S9 Fig. Log-linear plot of estimated transition rates with and without error as a function of clade age obtained from simulated character state transitions.** The figure displays the results of state transition rates estimated at various time points, using data simulated on two types of 64-taxon trees representing extremes in tree balance: pectinate (resembling "caterpillar" trees) and perfectly balanced trees, where diversity is evenly distributed between sister clades. The true rates were assumed to be constant over time, indicated by the red dashed line. The first column (a and c) illustrates rates with no errors in the datasets and constant rates over time, depicted by the blue trend line. The second column (b and d) shows rate estimates from the same data but with errors introduced by randomly selecting several taxa and changing their true state to the incorrect one before re-estimating the rate. These errors result in an artificial negative relationship between the transition rate and clade age. (TIF)

**S10 Fig. Log-linear plot of estimated transition rates with and without error as a function of sum of all branch lengths obtained from simulated character state transitions.** Note that this figure is similar to S9 Fig but replaces clade age on the vertical axis with the total time represented by a given tree (i.e., the sum of all branch lengths). Note the difference in scale for both axes. Pectinate trees are less affected by errors compared to balanced trees due to the greater overall time represented in the former. (TIF)

**S11 Fig. Simulation results for different scenarios of the *hmb* model described in the main text.** Each dot represents a simulated comparison. Within each subplot, the *x*-axis represents time and the y axis empirical rate, both on a log scale. The red line represents the true rate at that time under the model, the other solid lines the reconstructed rates under the best-fitting *hmb* model for each replicate dataset. The dashed line shows a linear fit to the points. The subgraphs are arranged in columns by generating models 1, 2, 3, and 4 and by dataset sizes of 500 or 5000 points and in rows by the standard deviation used in the simulation. Model 1 is a constant rate; models 2 and 3 are increasing and decreasing rates with increased time, respectively; model 4 is a sine wave with a periodicity of 26 million years. Note that with low measurement error, the clouds of points resemble the generating model (red line) but with increasing

amounts of noise the clouds begin to resemble the points found in the empirical datasets.
(TIF)

**S12 Fig. This shows multiple simulations and is arranged in the same way as S11 Fig.** Each purple or green line represents the *hmb* model fit for the generating rate from a different replicate; the true generating rate is in red. Note that the axes in this plot are not log transformed.
(TIF)

**S13 Fig. Plot arranged as in S12 Fig but showing the estimate of how much of the empirical rate stemmed from the hyperbolic process (equivalent to measurement error) as solid purple or green lines versus the median across the simulations (red line).** The match is not perfect, but *hmb* generally performs well despite in some cases (like model 4) not being able to match the complexity of the generating model.
(TIF)

**S1 Table. Summary of the results obtained from fitting a least-squares model to each empirical dataset and various iterations of our Yule simulated datasets.** Rows highlighted in bold indicate the models with the lowest AIC overall from the seven models fitted to each dataset. The values in parentheses beneath each of the model parameters denote the two ends of the 95% confidence interval obtained from our uncertainty analyses.
(XLSX)

**S1 Movie. An animated gif comparing the original empirical data and five different randomizations of the numerator with respect to the denominator.**
(GIF)

## Acknowledgments

This work presented here was substantially influenced by James Boyko's advocacy in challenging the artifactual nature of age rate scaling. Conversations with Dan Simberloff, Nina Fefferman, Paul Armsworth, Xingli Giam, Naomi O'Meara, Adam Siepielski, and Andy Alverson were also of substantial help in generating ideas and providing constructive feedback.

## Author Contributions

**Conceptualization:** Brian C. O'Meara, Jeremy M. Beaulieu.

**Data curation:** Brian C. O'Meara, Jeremy M. Beaulieu.

**Funding acquisition:** Brian C. O'Meara, Jeremy M. Beaulieu.

**Investigation:** Brian C. O'Meara, Jeremy M. Beaulieu.

**Methodology:** Brian C. O'Meara, Jeremy M. Beaulieu.

**Project administration:** Brian C. O'Meara, Jeremy M. Beaulieu.

**Software:** Brian C. O'Meara, Jeremy M. Beaulieu.

**Validation:** Brian C. O'Meara, Jeremy M. Beaulieu.

**Visualization:** Brian C. O'Meara, Jeremy M. Beaulieu.

**Writing – original draft:** Brian C. O'Meara, Jeremy M. Beaulieu.

**Writing – review & editing:** Brian C. O'Meara, Jeremy M. Beaulieu.

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
