## [Decision Letter · Decision Letter 0]

14 Aug 2024

Dear Dr. O'Meara,

Thank you very much for submitting your manuscript "Noise leads to the perceived increase in evolutionary rates over short time scales" for consideration at PLOS Computational Biology. As with all papers reviewed by the journal, your manuscript was reviewed by members of the editorial board and by several independent reviewers. The reviewers appreciated the attention to an important topic. Based on the reviews, we are likely to accept this manuscript for publication, providing that you modify the manuscript according to the review recommendations.

Please address the reviewers' criticisms point by point. Reviewer #1 requests an application of your model to other aspects in biology. Since this can be a rather expansive addition, I believe that this point would be addressed sufficiently by providing an example or two in the Discussion. 

Please address all other comments as requested by the reviewers.

Sincerely,

Iddo Friedberg, Ph.D.

Academic Editor

PLOS Computational Biology

Jason Papin

Editor-in-Chief

PLOS Computational Biology

Reviewer's Responses to Questions

**Comments to the Authors:**

Reviewer #1: This interesting paper explores aspects of a frequently examined topic: how to biological rates vary over time? The authors show that the so called 'pull to the recent' of such rates is frequently a statistical artifact of time-independent errors. They then propose a new LS approach that enables one to separate these effects, and provide software for its use; thereby enabling empiricists to evaluate such trends properly. I found the work to be well written and the problem, and solution, well explained. The Methods and SI were expansive and allow the more technical reader to follow, implement, and check the details.

My one suggestion is for broadening the connections between this advance and the extensive prior work on measurement error. In many biological disciplines, hierarchical models are used to account for measurement error while assessing primary patterns. One such phylogenetic approach was mentioned in the discussion, but their use in biology for this purpose is far more extensive. And indeed, one could consider the proposed model a form of hierarchical linear model, as the term 'h/t' is effectively the error component: (e2 -e1)/(t2 -t1). I would suggest stating this connection explicitly where the method is first derived (~pg. 7), with further comments on how measurement error is modeled in other types of biological settings in the Discussion. The authors may also wish to consider whether any broader applications of their method may be found.

Reviewer #2: Review of "Noise leads to the perceived increase in evolutionary rates over short time scales"

This manuscript examines the possible causes of a commonly observed pattern: that evolutionary rates appear to increase towards the present. The authors propose that such observations are wholly (or mostly) due to a statistical artifact, and then go on to show that such an artifact mimics real data. In addition, they provide a statistical method for disentangling such bias from real patterns.

I really liked this paper. It was generally clearly laid out and clearly explained, with the admission of limitations where appropriate. While I think it might annoy quite a few people, this is an important area and the truth (or not) about time-varying rates is a crucial question to address for the field. I also think the ideas are novel and timely.

That said, I have some major and minor comments that I hope will help to improve the presentation:

1. Figures need better explanation. There is a lot going on in the figures, but there is not a lot of explanation. One general request: instead of pointing to, e.g., “Figure 1,” point to a specific panel (or column or row) of Figure 1 and be specific.

In Figure 1 in particular, there are a lot of colors and a lot of different things going on in each panel. Please give more explanation. For instance, on line 229 the text states that the relationship is “caused by the hyperbolic component (Fig. 1).” What should I be looking for in Figure 1 that shows this? Is it that there is dark pink overlapping the black lines toward the present? There is little explanation of all the colors and how they are fit in the first column of this figure. (Also, what are the green dashed lines?) Is there some explanation for the different patterns of colored lines in the third column between simulated and real data?

Similarly, Figure 2 is underexplained. Why is the OR value lower than the OO value for the Ho et al. dataset? Does this mean something? What do all the p-values on the comparisons represent, and why are they significant even in the simulated data? Panel labels might also help here.

2. Measurement error. I really liked the idea that measurement error is constant, so that it has a larger effect on more recent rates. That said, I think a clearer explanation of what constitutes “error” would be extremely helpful.

Of course, I do understand measurement error narrowly defined: using calipers to get a height is not always precise, or sequencing data contains errors. But the field of comparative methods seems to also have a broader notion of measurement error that is not always clear. I am hoping the manuscript can clarify this at least a bit.

For example, the text says: “Measurements of x_i often reflect the mean for a species or even a measurement of a single individual and, therefore, represent an estimate of the true value” (lines 126-127). What exactly is the “true value” imagined to be here? If I were estimating nucleotide rates I would usually only have a single individual at the tip of my tree, and in fact I believe that this is what I would correctly be modeling.

But my impression is that much of the comparative methods literature believes that the species mean should be the data at the tips of the tree, and this is the sense in which even a small sample of individuals might give an erroneous estimate of this mean. (This is clearly what the discussion on lines 355-357 implies.) But does this type of error have the same effect on comparative methods? If we are modeling the species means, does this imply that there is a different type of error than if we are modeling single individuals at the tips? What about the effects of plasticity? I imagine this is also “error” here?

I think there might also be another type of “error” in lots of analyses: violations of the underlying assumptions. For example, even though most analyses of substitution rates assume d=2tk (the genetic distance is twice the time elapsed since the MRCA multiplied by the substitution rate), the more accurate expectation is d=2tk+theta, where theta is the amount of ancestral polymorphism (Gillespie and Langley 1979, JME). The “error” in ignoring theta is of course small far back in time, but plays a more larger role more recently. (My intuition is that the same thing affects quantitative traits.) But of course here there is not measurement error sensu stricto, but instead an unmodeled process that is ~constant through time.

All in all, a clearer view of what types of errors might cause the patterns observed here would be very useful, as would a discussion of the solutions to modeling error (e.g. lines 355-357) and whether there are solutions applicable to all of these types of error.

3. Additional literature. I appreciated that this manuscript had many citations to the classic literature on the statistical artifacts. There are of course many other papers in this area, and below I list a few that might be relevant. There is no requirement that any of these be cited, but I thought they might be helpful.

-Brett (2004, Oikos) examined the effect of the amount of error on spurious correlations. This paper might also be used as a starting place to examine the magnitude of error needed to drive the effects here.

-Kronmal (1993, Journal of the Royal Statistical Society A) and Jackson and Somers (1991, Oecologia) both discuss modeling rates using interaction terms in linear models (or close variants of this approach) to avoid some of the problems that are also shown here.

-Jackson and Somers (1991, Oecologia) also promote the use of randomization if ratios have to be used.

4. Minor comments

-Line 26: what is “conservation” referring to here?

-Line 138: remove comma after “hand-side”?

-Lines 199-204: what do the observations about heterogeneity mean? This was unclear.

-Line 213: this is the first place the phrase “hmb model” is used. Please define earlier.

-Lines 360-363: there are a number of methods for dealing with errors in molecular measurements in comparative models. See, for instance, Kuhner and McGill (2014, Genetics) and Han et al. (2013, MBE).

-Figure 1: how can there be times less than 0 on the x-axis? (Second and third rows)

Reviewer #3: O’Meara and Beaulieu show that the well-known apparent trend of higher evolutionary rates towards the present or on short time scales is due to errors in the measure of traits whose evolution we study. The relation between "a noisy numerator divided by time versus time" is surprisingly efficient at explaining observed patterns.

This paper was truly a pleasure to read. First, it is intellectually statisfying that a pattern which looked like a bias is finally explained convincingly as one, and a simple one. It is a strength of very good science that the answer seems obvious in retrospect, and indeed I was surprised to not be able to find prior publications showing the results presented here. Second, the authors provide all the verifications and checks which I would expect, and I was constantly pleasantly surprised to find that additional analyses that I intended to ask for had in fact been done. Third, it is rare that a paper with such a methodological focus is so pleasant to read yet so clear in its message and content.

Considering all of this, I only have a few comments, to the discretion of the editor:

Major comments:

p. 14, the authors mention briefly that "Estimates of time are biased and uncertain." Given the rich literature on the conflicts between paleontological and molecular clock estimates of evolutionary time, and their respective biases, I would appreciate some more discussion of the impact of these biases and uncertainties.

Some pointers to this literature:

https://doi.org/10.1144/jgs2021-107

https://doi.org/10.1371/journal.pbio.3001998

https://doi.org/10.1016/j.cub.2023.06.016

https://doi.org/10.1002/bies.201600120

https://doi.org/10.1016/j.tig.2020.06.002

p. 23, the percentile calculation for the pvalues is uncharacteristically complicated to understand, and I am afraid that it will be confusing to readers relative to the usual meaning of "pvalue". Moreover, I am not convinced that this test adds to the strength of the results.

Minor comments:

line 234-236, it isn't immediately obvious which simulation approach was used, please clarify. This whole paragraph would be easier to follow if subfigures were referenced where relevant (e.g., Fig 1A).

line 452, there seems to be something missing in this sentence "and AIC Adding uncertainty in our plots section below)."

line 578, please clarify that you mean the online supplemental documents at Zenodo.

**Have the authors made all data and (if applicable) computational code underlying the findings in their manuscript fully available?**

Reviewer #1: Yes

Reviewer #2: Yes

Reviewer #3: Yes

PLOS authors have the option to publish the peer review history of their article (what does this mean?). If published, this will include your full peer review and any attached files.

Reviewer #1: No

Reviewer #2: No

Reviewer #3: No

Figure Files:

Data Requirements:

Reproducibility:

References:

---

## [Editor Report · Decision Letter 1]

4 Sep 2024

Dear Dr. O'Meara,

We are pleased to inform you that your manuscript 'Noise leads to the perceived increase in evolutionary rates over short time scales' has been provisionally accepted for publication in PLOS Computational Biology.

Best regards,

Iddo Friedberg, Ph.D.

Academic Editor

PLOS Computational Biology

Jason Papin

Editor-in-Chief

PLOS Computational Biology

---

## [Editor Report · Acceptance letter]

9 Sep 2024

PCOMPBIOL-D-24-01129R1 

Noise leads to the perceived increase in evolutionary rates over short time scales

Dear Dr O'Meara,

I am pleased to inform you that your manuscript has been formally accepted for publication in PLOS Computational Biology. Your manuscript is now with our production department and you will be notified of the publication date in due course.

With kind regards,

Zsofia Freund
